

# Environmental and taxonomic controls of carbon and oxygen stable isotope composition in *Sphagnum* across broad climatic and geographic ranges

Gustaf Granath[1], Håkan Rydin[1], Jennifer L. Baltzer[2], Fia Bengtsson[1], Nicholas Boncek[3], Luca Bragazza[4,5,6], Zhao-Jun Bu[7,8], Simon J. M. Caporn[9], Ellen Dorrepaal[10], Olga Galanina[11], Mariusz Gałka[12], Anna Ganeva[13], David P. Gillikin[14], Irina Goia[15], Nadezhda Goncharova[16], Michal Hájek[17], Akira Haraguchi[18], Lorna I. Harris[19], Elyn Humphreys[20], Martin Jiroušek[21, 22], Katarzyna Kajukało[12], Edgar Karofeld[23], Natalia G. Koronatova[24], Natalia P. Kosykh[24], Mariusz Lamentowicz[12], Elena Lapshina[25], Juul Limpens[26], Maiju Linkosalmi[27], Jin-Ze Ma[7,8], Marguerite Mauritz[28], Tariq M. Munir[29, 30], Susan Natali[31], Rayna Natcheva[13], Maria Noskova†, Richard J. Payne[32, 33], Kyle Pilkington[3], Sean Robinson[34], Bjorn J. M. Robroek[35], Line Rochefort[36], David Singer[37], Hans K. Stenøien[38], Eeva-Stiina Tuittila[39], Kai Vellak[23], Anouk Verheyden[14], James Michael Waddington[19], Steven K. Rice[3]

[1]Department Ecology and Genetics, Uppsala University, Norbyvägen 18D, Uppsala, Sweden
[2]Biology Department, Wilfrid Laurier University, 75 University Ave. W., Waterloo, ON, N2L 3C5, Canada
[3]Department of Biological Sciences, Union College, Schenectady, NY, US
[4]Department of Life Science and Biotechnologies, University of Ferrara, Corso Ercole I d'Este 32, I-44121 Ferrara, Italy
[5]Swiss Federal Institute for Forest, Snow and Landscape Research, WSL Site Lausanne, Station 2, 1015 Lausanne, Switzerland
[6]Ecole Polytechnique Fédérale de Lausanne EPFL, School of Architecture, Civil and Environmental Engineering ENAC, Laboratory of ecological systems ECOS, Station 2, 1015 Lausanne, Switzerland
[7]Institute for Peat and Mire Research, Northeast Normal University, State Environmental Protection Key Laboratory of Wetland Ecology and Vegetation Restoration, 5268 Renmin St., Changchun 130024, China
[8]Jilin Provincial Key Laboratory for Wetland Ecological Processes and Environmental Change in the Changbai Mountains, 5268 Renmin St., Changchun 130024, China
[9]School of Science and the Environment, Division of Biology and Conservation Ecology, Manchester Metropolitan University, Manchester, M1 5GD, UK
[10]Climate Impacts Research Centre, Dept. of Ecology and Environmental Science, Umeå University, 98107 Abisko, Sweden
[11]Institute of Earth Sciences, St. Petersburg State University, Universitetskaya nab., 7-9, Russia, 199034, St.Petersburg, Russia
[12]Department of Biogeography and Paleoecology, Adam Mickiewicz University in Poznan, Bogumiła Krygowskiego 10, 61-680 Poznan, Polen
[13]Institute of Biodiversity and Ecosystem Research, Bulgarian Academy of Sciences, 2 Yurii Gagarin Str., 1113 Sofia, Bulgaria
[14]Department of Geology, Union College, Schenectady, NY, USA
[15]Babeș-Bolyai University, Faculty of Biology and Geology, Department of Taxonomy and Ecology, 42 Republicii Street, RO-400015, Cluj Napoca, Romania
[16]Institute of Biology of Komi Scientific Centre of the Ural Branch of the Russian Academy of Science, Russia
[17]Department of Botany and Zoology, Faculty of Science, Masaryk University, Kotlarska 2, CZ-61137, Brno, Czech Republic
[18]Department of Biology, The University of Kitakyushu, Kitakyushu 8080135, Japan
[19]School of Geography and Earth Sciences, McMaster University, 1280 Main Street West, Hamilton, ON, L8S 4K1, Canada
[20]Department of Geography and Environmental Studies, Carleton University, Ottawa, Canada



[21]Department of Botany and Zoology, Faculty of Science, Masaryk University, Kotlarska 2, CZ-61137, Brno, Czech Republic

[22]Department of Plant Biology, Faculty of AgriSciences, Mendel University in Brno, Zemedelska 1, CZ-61300, Brno, Czech Republic

[23]University of Tartu, Institute of Ecology and Earth Sciences, Lai st 40, Tartu 51005, Estonia

[24]Laboratory of Biogeocenology, Institute of Soil Science and Agrochemistry, Siberian Branch of Russian Academy of Sciences, Ak. Lavrent'ev ave., 8/2, Novosibirsk, 630090, Russia

[25]Yugra State University, Chekhova str, 16, Khanty-Mansiysk, 628012, Russia

[26]Plant Ecology and Nature conservation group, Wageningen University, Droevendaalse steeg 3a, 6708 PD Wageningen, the Netherlands

[27]Finnish Meteorological Institute, Erik Palménin aukio 1, FI-00560 Helsinki, Finland

[28]Center for Ecosystem Science and Society (Ecoss), Department of Biological Sciences, Northern Arizona University, POBox 5620, Flagstaff, AZ 86011, USA

[29]Department of Geography, University of Calgary, 2500 University Dr. NW, Calgary, AB, T2N 1N4, Canada

[30]Department of Geology, St. Mary's University, Calgary, AB, T2X 1Z4, Canada

[31]Woods Hole Research Center, 149 Woods Hole Road, Falmouth MA 02540, USA

[32]Environment, University of York, York YO105DD, UK

[33]Penza State University, Krasnaya str., 40, 440026 Penza, Russia

[34]Department of Biology, SUNY-Oneonta, Oneonta, NY, USA

[35]Biological Sciences, University of Southampton, Southampton SO17 1BJ, UK

[36]Department of Plant Science and Center for Northern Studies, Laval University, QC, Canada

[37]Laboratory of Soil Biodiversity, Institute of Biology, University of Neuchâtel, Rue Emile-Argand 11, CH-2000 Neuchâtel, Switzerland

[38]NTNU University Museum, Norwegian University of Science and Technology, Erling Skakkes gate 47, N-7491 Trondheim, Norway

[39]School of Forest Sciences, University of Eastern Finland, B.O. Box 111, FIN-80110 Joensuu, Finland

[†]deceased, 27 August 2017


*Correspondence to*: Gustaf Granath (gustaf.granath@gmail.com)

**Abstract.** Rain-fed peatlands are dominated by peat mosses (*Sphagnum* sp.), which for their growth depend on elements from the atmosphere. As the isotopic composition of carbon ($^{12,13}C$) and oxygen ($^{16,18}O$) of these *Sphagnum* mosses are affected by environmental conditions, the dead *Sphagnum* tissue accumulated in peat constitutes a potential long-term

archive that can be used for climate reconstruction. However, there is a lack of adequate understanding of how isotope values are influenced by environmental conditions, which restricts their current use as environmental and palaeoenvironmental indicators. Here we tested **(i)** to what extent C and O isotopic variation in living tissue of *Sphagnum* is species-specific and associated with local hydrological gradients, climatic gradients (evapotranspiration, temperature, precipitation), and elevation; **(ii)** if the C isotopic signature can be a proxy for net primary productivity (NPP) of *Sphagnum*;

and **(iii)** to what extent *Sphagnum* tissue $\delta^{18}O$ tracks the $\delta^{18}O$ isotope signature of precipitation. In total, we analysed 337 samples from 93 sites across North America and Eurasia using two important peat-forming *Sphagnum* species (*S. magellanicum, S. fuscum*) common to the Holartic realm. There were differences in $\delta^{13}C$ values between species. For *S. magellanicum* $\delta^{13}C$ decreased with increasing height above the water table (HWT, $R^2=17\%$) and was positively correlated to productivity ($R^2=7\%$). Together these two variables explained 46% of the between-site variation in $\delta^{13}C$ values. For *S.*



*fuscum*, productivity was the only significant predictor of δ¹³C (total $R^2$=6%). For δ¹⁸O values, ca. 90% of the variation was found between sites. Globally-modelled annual δ¹⁸O values in precipitation explained 69% of the between-site variation in tissue δ¹⁸O. *S. magellanicum* showed lower δ¹⁸O enrichment than *S. fuscum* (-0.83‰ lower) . Elevation and climatic variables were weak predictors of tissue δ¹⁸O values after controlling for δ¹⁸O values of the precipitation. To summarise, our study provides evidence for **(a)** good predictability of tissue δ¹⁸O values from modelled annual δ¹⁸O values in precipitation, and **(b)** the possibility to relate tissue δ¹³C values to HWT and NPP, but this appears to be species-dependent. These results suggest that isotope composition can be used at a large scale for climatic reconstructions but that such models should be species-specific.

## 1 introduction

Peatlands in temperate, boreal and arctic regions form large reservoirs of carbon, which are vulnerable to release under expected changes in global climate and land management (Rydin and Jeglum 2013, Loisel et al. 2014). Because peat decomposes slowly and gradually accumulates, it preserves information on past peatland ecosystem dynamics and responses to allogenic and autogenic forcing. Palaeoenvironmental studies of peat may, therefore, help anticipate future response of these globally important ecosystems to climate change (Loader et al. 2016). Past climate and local hydrology can be estimated using a variety of biotic and biogeochemical proxies, including the δ¹³C and δ¹⁸O values of organic material.  (e.g. van der Knaap 2011, Royles et al. 2016, Lamentowicz et al. 2015). However, the environmental (e.g. climate) and biotic (e.g. species identity) controls of isotope differentiation in peatland-dwelling plants is still poorly understood, and current assumptions regarding these controlling factors are yet to be tested at larger spatial scales.

*Sphagnum* mosses are the most dominant peat-forming plant group in acidic peatlands. The composition of stable isotopes of carbon and oxygen in their tissues is affected by different environmental conditions, operating through their impact on fractionation processes. When not submerged, carbon isotope signals in bulk tissues or components such as cellulose depend mainly on the $[CO_2]$ in the chloroplast ($[CO_2]_c$), which alters isotope discrimination during biochemical fixation of $CO_2$ and by fractionation caused by diffusion to the chloroplast (Farquhar et al. 1989, O'Leary 1988). In mosses, $[CO_2]_c$ has been shown to be determined by temperature, light availability, $CO_2$ partial pressure and, most importantly, plant water status (Finsinger et al. 2013, van der Knaap et al. 2011, Ménot and Burns 2001, Ménot-Combes et al. 2004, Royles et al. 2014, Skrzypek et al. 2007a, Kaislahti Tillman et al. 2013). When wet, external water films on leaf surfaces impede diffusion and $[CO_2]_c$ is lowered (Rice and Giles 1996, Rice 2000, Williams and Flanagan 1996); consequently, the proportion of fixed ¹³C increases due to internal drawdown of the preferred isotope ¹²C. When submerged, assimilation of methane-derived $CO_2$ can alter $[CO_2]$ and also the C isotopic composition of *Sphagnum* (Raghoebarsing et al. 2005). Despite many detailed studies, there remains-uncertainty about how the multiple controls on ¹³C isotope values combine to determine isotopic composition,





and how universal the proposed mechanisms are on a global scale. This uncertainty currently restricts the utility of C isotope signals as a palaeoclimatic/palaeoenvironmental indicator in peatlands (Loader et al. 2016).

Oxygen isotope values in moss tissues depend on the isotopic composition of the water sources, enrichment associated with evaporation from the moss surface and biochemical fractionation (Dawson et al. 2002). Once on the plant, $^{18}$O present in water equilibrates with that in atmospheric $CO_2$ prior to fixation as well as being incorporated directly during hydrolysis reactions, especially during the initial stages of carbon fixation (Gessler et al. 2014, Sternberg et al. 2006). Hence, variation in tissue oxygen isotopes reflect environmental conditions that control source water (rainfall, snowfall, groundwater) as well as fractionation caused by evaporation prior to fixation which is controlled by micrometeorological conditions (mainly

temperature, relative humidity and incident energy) (Daley et al. 2010, Moschen et al. 2009, Royles et al. 2013, Kaislahti Tillman et al. 2010). Oxygen isotope composition has, therefore, been used to reconstruct climatic conditions and to infer the dominant water source in peatlands (Aravena and Warner 1992, Ellis and Rochefort 2006, van der Knaap et al. 2011). Ongoing measurements of oxygen isotopes in precipitation across the globe (Bowen 2010, IAEA/WMO 2015) have generated models that predict spatial patterns in oxygen isotope composition of precipitation based on temperature,

elevation, atmospheric residence time and circulation patterns (e.g. Bowen 2010). Once isotopic composition of the source water is accounted for, variation in moss tissue isotopic values should be largely determined by fractionation that accompanies evaporation from the surface of plants. How well oxygen isotopes in *Sphagnum* tissues reflect atmospheric water or plant surface water depend on local weather conditions such as precipitation, air temperature and humidity. For example, Bilali et al. (2013) suggest that oxygen isotopes in *Sphagnum* mosses from maritime bogs will track variation in

precipitation patterns whereas isotopic values in continental habitats will be more dependent on summer temperature, as temperature and humidity are more variable in those regions. At local scales, oxygen isotope values also vary as a function of temperature and humidity. Aravena and Warner (1992) found differences that correspond with changes in microtopography. Elevated microsites (hummocks) were enriched in $^{18}$O, which they ascribed to higher evaporation compared to that of neighbouring wet depressions (hollows). However, as with $^{13}$C, there remains uncertainty in how $^{18}$O

signatures relate to environmental factors and species identity, and to what extent global $^{18}$O patterns in precipitation dominate over local processes.

Stable isotopes can also serve as indicators of primary productivity (NPP) (Rice and Giles 1996, Williams and Flanagan 1996, Rice 2000). However, few studies have explored these relationships in the field. In a multispecies comparison of peat

mosses, Rice (2000) found that plants with higher relative growth rates had lower discrimination against $^{13}$C and therefore were more enriched in $^{13}$C. This was attributed to the local environment, with fast growing plants of wetter microhabitats having thicker water films that inhibits $CO^2$ diffusion into the plant, and to species-specific differences in maximum rates of photosynthesis. Both factors would reduce internal $[CO_2]$ and thereby lower discrimination. In line with this, a warming experiment by Deane-Coe et al. (2015) reported a positive relationship between moss net primary productivity (NPP) and



$\delta^{13}$C values for tundra mosses (*Dicranum*, *Pleurozium*, *Sphagnum*). Clearly, carbon isotope values show promise as indicators of peat moss contemporary growth, and potentially as a NPP proxy in paleoecological studies. This could be particularly valuable to differentiate productivity and decomposition controls in long-term carbon accumulation studies. To date, we are not aware of attempts to explore the robustness of these relationships across large spatial scales.

Together, tissue carbon and oxygen isotope composition are controlled both by environmental factors at micro- and macro-scales, and by species-specific differences that relate to water balance and carbon dynamics in peat mosses. Paleoecological studies rely on such environment—isotope relationships for environmental reconstructions (Ellis and Rochefort 2006, van der Knaap et al. 2011). The underlying mechanisms are, however, rarely fully explored using known environmental gradients (but see Ménot and Burns 2001 for an example), or only tested across narrow bands of environmental variation,

often with sets of correlated environmental factors (Loader et al. 2016). Moreover, interactions with biotic factors such as species identity have received little attention despite the large variations in *Sphagnum* species dominance commonly observed down peat cores (e.g. Ménot and Burns 2001). Here we aim to provide a robust, cross-scale evaluation of how environmental factors and species identity influence the C and O isotope composition of *Sphagnum* using two common and widely distributed peat-forming species (*S. magellanicum* and *S. fuscum*) that are primarily rain-fed. To achieve this, we

performed a unprecedented large sampling campaign across the Holarctic realm.

Specifically, we **(i)** investigated relationships between C and O isotope values and factors known to influence plant water availability (height above the water table - HWT, temperature, evaporation and precipitation) and $CO_2$ partial pressure (elevation), and tested if their effects were modified by species identity; **(ii)** tested the prediction that *Sphagnum* tissue $\delta^{13}$C

values are associated with NPP; **(iii)** tested if tissue $\delta^{18}$O in rain-fed Sphagna is predicted by the $\delta^{18}$O isotope signature in precipitation but modified by negative relationships with precipitation and positive ones with temperature/evaporation. Across these objectives we examined how C and O isotope values varied with scale (within-peatland *vs* among peatlands) and to what extent HWT and NPP could explain variation within and between peatlands.

## 2 Materials and Methods

### 2.1 Study species and collection sites

Our study focused on two common peat-forming *Sphagnum* species, *S. fuscum* (Schimp.) H. Klinggr. (circumpolar distribution) and *S. magellanicum* Brid. (cosmopolitan distribution). In general, these species are confined to primarily rain-fed peatlands (bogs), and described as hummock (*S. fuscum*) and lawn (*S. magellanicum*) species. However, *S. magellanicum* is a species with a very broad niche and found in a range of habitats with varying degrees of groundwater

influence (Flatberg 2013). These species are easy to identify but recent research has shown that the dark European morph of *S. fuscum* is conspecific to the North American *S. beothuk* (Kyrkjeeide et al. 2015), and *S. magellanicum* has been shown to





consist of two genetically diverged morphotypes (Kyrkjeeide et al. 2016). Unpublished genetic data suggest that samples collected in our study consist of both *S. magellanicum* morphs (approximately 50/50) and possibly one or two samples of *S. beothuk* (Pers. comm. N. Yousefi). Hence, we here treat our species as aggregates (i.e. species collectiva), *S.fuscum* coll.

and *S. magellanicum* coll..

The two species were sampled across the Holarctic region at a total of 93 sites (Figure 1; Supplemental Table S1) at the end of the growing season. To make comparisons between species and sites possible, we focused on habitats where both species can be found and have low influence of surrounding groundwater. Thus, we only sampled bogs (including a few poor fens

with ombrotrophic character) and open (no tree canopy) habitats. Sampling was conducted mainly during 2013, but a few sites were sampled at a similar time of year in 2014. At each site two patches (minimum 10 m apart) for each species were sampled (except for 11 sites that contained only one patch for one species). At each sampling patch we recorded moss growth, HWT (height above the water table) and GPS coordinates, and collected a moss sample (78 cm$^2$ and 5 cm deep) at the end of the growing season (September to November depending on location). Moss samples were dried (24 hrs at 60-65

°C) within 72 h, or alternatively immediately frozen and later thawed and dried. The apical part (the capitula, top 1 cm) of the dried plant shoots was used for isotope analysis, while the stem section was used for bulk density estimation to calculate moss NPP.

**2.2 Isotope determination**

Ten capitula from each patch were selected and finely chopped with a single-edge razor by hand and mixed. Standard

deviations of repeated samples were 0.6 and 0.7 ‰ for $\delta^{13}$C and $\delta^{18}$O, respectively. Approximately 0.5 mg dry sample was packed in tin cups for $\delta^{13}$C analyses, and ~0.2 mg in silver cups for $\delta^{18}$O analyses. Samples were analyzed at Union College (Schenectady, NY, USA) using a Thermo Delta Advantage mass spectrometer in continuous flow mode connected (via a ConFlo IV) to a Costech Elemental Analyzer for $\delta^{13}$C analysis or a Thermo TC/EA for $\delta^{18}$O analyses. Isotope values are presented as $1000 \times (R_{sample}/R_{standard}-1)$, where $R_{sample}$ and $R_{standard}$ are the ratios of heavy to light isotopes (e.g., $^{13}$C/$^{12}$C) andare

referenced to VPDB and VSMOW for C and O, respectively. Carbon isotope data were corrected using sucrose (IAEA-CH-6, -10.449‰), acetanilide (in house, -37.07‰), and caffeine (IAEA-600, -27.771‰). Oxygen isotope data were corrected using sucrose (IAEA-CH-6, 36.4‰), cellulose (IAEA-C3, 31.9‰) and caffeine (IAEA-600, -3.5‰) with values from Hunsinger et al. (2010). Oxygen isotope standardization was further checked with the whole wood standards USGS54 and USGS56. The combined instrument uncertainty for $\delta^{13}$C (VPDB) is < 0.1‰ based on the in-house acetanilide standard and <

0.5‰ for $\delta^{18}$O (VSMOW) based on the cellulose standard (IAEA-C3).

We performed isotope analyses on whole-plant tissue rather than on cellulose extracts. In living *Sphagnum* samples, there is strong linear relationship between the isotopic composition of these two components for both $\delta^{13}$C (R$^2$ values 0.89-0.96; Kaislahti Tillman et al. 2010, Ménot and Burns 2001, Skrzypek et al. 2007b) and for $\delta^{18}$O (R$^2$ values 0.53-0.69; Kaislahti



Tillman et al. 2010, Jones et al. 2014). Focussing on whole-plant tissue allowed us to analyze a higher number of samples for this study, allowing larger numbers of sites and more replication.

**2.3 Environmental variables**

The modelled $\delta^{18}O$ signal in meteoric water (precipitation) (Bowen and Wilkinson 2002) was obtained from http://www.waterisotopes.org as annual and monthly isotope ratio estimates at 10' resolution. These global estimates have 220   shown to be highly accurate ($R^2 = 0.76$ for mean annual $\delta^{18}O$ in precipitation) and are based on absolute latitude and elevation and account for regional effects on atmospheric circulation patterns (for details see Bowen 2010, IAEA/WMO 2015, Bowen 2017). To test which temporal period of $\delta^{18}O$ values in precipitation showed the highest correlation with tissue $\delta^{18}O$ values, we calculated annual (Jan-Dec), growing season (May-Oct), winter-spring (Jan-April) mean isotope ratio. We calculated both unweighted means and weighted against precipitation for each month. Monthly precipitation 225   (PRECTOTCORR), land evapotranspiration (EVLAND) and surface air temperature (TLML) for each site and year of sampling (2013 or 2014) were retrieved from the NASA GESDISC data archive, land surface and flux diagnostics products (M2T1NXLND, M2TMNXFLX; resolution longitude 0.667°, latitude 0.5°; Global Modeling and Assimilation Office 2015ab). Total precipitation and evapotranspiration (ET), and mean temperature, from April to October were used as predictors in the statistical models. As ET can be compensated for by precipitation, we used the ET/P quotient as a predictor 230   for the effect of water loss. A high value (>1) indicates a net loss of water to the atmosphere. Site altitude was retrieved from a global database using the R package *elevatr* (ver 0.1-2, Hollister and Shah 2017).

The distance from the moss surface to the water table (height above the water table, HWT), was measured using water wells (commonly a PVC pipe, 2–5 cm in diameter and slotted or perforated along the sides) with a "plumper" (a cylinder on a 235   string that makes a 'plump' sound when it hits the water surface) or a "bubbler" (a narrow tube that makes bubbles when it hits the water surface while the user blows in it).

**2.4 Moss growth**

Moss growth (or productivity, NPP) was measured with a modified version of the cranked wire method (see Clymo 1970 and Rydin and Jeglum 2013 for details), with bristles from a paint brush spirally attached to a wire. These 'brush wires' were 240   inserted in the moss layer with the end of the wire protruding above the surface. Height increment (i.e. vertical growth) was measured over the growing season as the change in distance (to nearest mm) between the moss layer and the top of the wire. A minimum of three wires were inserted within a 1 x 1 m uniform area (same microhabitat, vegetation and general structure). To determine moss bulk density ($g\ m^{-3}$) we dried (24 hrs at 60-65 °C) the top 30 mm of the stems in our collected core (see Sect. 2.1). Biomass growth on an area basis ($g\ m^{-2}\ yr^{-1}$) was calculated as height increment × bulk density.


### 2.5 Statistical analyses

To test and quantify the influence of environmental variables and species identity on isotope composition, we used linear mixed models in R (R core team 2016), employing the R package *lme4* ver 1.1-12 (Bates et al. 2015). Site dependence (i.e. multiple samples from the same site) was accounted for by adding site as a random factor. For tissue $\delta^{13}$C, we first fitted two separate models to test the independent effect of HWT, NPP and species identity (*S. fuscum* and *S. magellanicum*), and if the HWT or NPP effect varied between species by fitting a species interaction term. To test the explanatory power of environmental variables (ET/P, precipitation, temperature, elevation) we first build a base model with HWT and NPP as they are identified as the main predictors in literature. For simplicity we removed negligible interactions from this model. Each environmental variable and their interaction with species was then tested against the base model. For tissue $\delta^{18}$O, we first explored which temporal period of modelled $\delta^{18}O_{precip}$ (annual, growing season, winter-spring) had the highest explanatory power and if the relationship varied between species. The identified best model was then used as base model to separately test each environmental variable (HWT, ET/P, precipitation, temperature) and its interaction with species.

The proportion of variance explained by the predictors was calculated at the site level (Gelman and Hill 2007) or as marginal $R^2$ (Nakagawa and Schielzeth 2013; R package *piecewiseSEM* ver 1.1-4, Lefcheck 2015). Although our study focus on explained variance by predictors, we also performed statistical tests of predictors and their interactions using type-2 (main effects tested after all the others in the model but without the interaction term) *F*-tests, applying Kenward-Roger adjustments of the degrees of freedom, as implemented in the *car* package (ver. 2.1-3, Fox and Weisberg 2011). Standard model checking was performed (e.g., residual analyses and distribution of random effects), to ensure compliance with model assumptions. Covariances among predictors were small (r < 0.15) or moderate (r = 0.40-0.50 among ET/P, precipitation and temperature) and this multicollinearity had minor impact on model estimates.

## 3 Results

### 3.1 $\delta^{13}$C signal

Variation in *Sphagnum* tissue $\delta^{13}$C values was marginally greater within sites than between sites (Table 1). HWT predicted the $\delta^{13}$C values, but the relationship differed between the two species (Table 2, Figure 2). Although $\delta^{13}$C values decreased with increasing HWT for both species, the slope was less steep for *S. fuscum* and this species had slightly higher $\delta^{13}$C values overall. In separate models for the two species, HWT for *S. fuscum* had near zero explanatory power, while for *S. magellanicum* HWT explained 33% of the between-site variation, and 17% of the total variance (i.e., marginal $R^2$).

Measured $\delta^{13}$C values were related to moss productivity (NPP), and $\delta^{13}$C values increased by 0.0023‰ (SE: 0.00048) for each mg biomass produced per $m^2$. NPP explained 11% of the between-site variation in $\delta^{13}$C and 7% of the total variation.



HWT and NPP, explained 48% of the between-site variation of $\delta^{13}C$ in *S. magellanicum*, and 24% of the total variation. Corresponding values for *S. fuscum* were 6% and 7%, respectively. Of the additional environmental variables tested, we found weak evidence that ET/P and temperature were positively correlated with $\delta^{13}C$, but only for *S. magellanicum* (Table 2).

## 3.2 $\delta^{18}O$ signal

*Sphagnum* tissue $\delta^{18}O$ values varied more among sites than within sites, and at similar magnitude and proportions for both species (Table 1). Tissue $\delta^{18}O$ values were predicted by the spatially explicit estimates of $\delta^{18}O$ values isotope signature in precipitation (Figure 3, Table 3). Annual mean $\delta^{18}O_{precip}$ explained 69% of the variation in $\delta^{18}O_{tissue}$ among sites. This was similar to mean winter-spring (Jan-Apr) $\delta^{18}O_{precip}$ values (75% explained), but higher than growing season (Apr-Sep) $\delta^{18}O_{precip}$
(58%). Using precipitation-weighted $\delta^{18}O_{precip}$ values resulted in lower percentages of explained variance for all three time periods ($R^2_{site}$: annual 52%, Jan-Apr 65%, Apr-Sep 52% ).

*S. magellanicum* had consistently lower $\delta^{18}O$ values than *S.fuscum* (-0.83‰), but both species had a similar relationship between tissue $\delta^{18}O$ and $\delta^{18}O_{precip}$ (Figure 3, Table 3).

Height above the water table (HWT) at the end of the growing season was, on average, 11 cm lower in *S. magellanicum* patches (= wetter habitat) compared to *S. fuscum* (HWT = 33 cm) patches ($F_{1,224}$ = 131.9, P < 0.0001). However, we found only very weak support for the hypothesis that HWT predicts tissue $\delta^{18}O$ values, as HWT explained <1% of the $\delta^{18}O$ variation (Table 2). There was negligible influence of the additional environmental variables on $\delta^{18}O$ values (Table 2). ET/P was associated with higher $\delta^{18}O$ values in *S. magellanicum* and lower in *S. fuscum* (but not different from zero effect), while
increasing temperature was weakly associated with overall lower $\delta^{18}O$ values.

# 4 Discussion

## 4.1 Stable carbon isotope discrimination in *Sphagnum*

Our data were consistent with the hypothesis that moss growing closer to the water table (low HWT) has reduced carbon isotope fractionation, leading to greater fixation of $^{13}CO_2$ and more $^{13}C$ enriched tissue (Rice and Giles 1996, Williams and
Flanagan 1996). Given that the water table position was measured in different places at different times and all are one-time measurements, this result is remarkably robust. For example refixation of $^{12}C$ enriched substrate-derived $CO_2$ in living Sphagna (Raghoebarsing et al. 2005) can potentially contribute to within-site variation in $\delta^{13}C$. Interestingly, the strength of this relationship differed in the two species, with *S. magellanicum* exhibiting a greater reduction in $\delta^{13}C$ in response to drier conditions (high HWT) than *S. fuscum*. The weaker effect of HWT on $\delta^{13}C$ values in *S. fuscum* is likely a consequence of
limited fluctuation in tissue water content as this species is well known to store abundant water within capillary spaces and



resist drying (Rydin 1985), thus maintaining the waterfilm that hampers fractionation. Loader et al. (2016) reported a similar slope estimate for *S. magellanicum* in a single peatland and several studies have confirmed effects of contrasting microtopography (i.e. hummock—hollow differences) using multi-species comparisons (Price et al. 1997, Loisel et al. 2009, Markel et al. 2010). As such, our results suggest that species-specific differences in carbon isotope discrimination in

*Sphagnum* are related to water retention capacity and, consequently, become more apparent under drier conditions. This supports the results of previous, smaller-scale studies (Rice 2000).

The influence of species identity on the relationship between $\delta^{13}C$ values and water table position has important implications for palaeoenvironmental reconstructions based on $\delta^{13}C$ values. The relationship between $\delta^{13}C$ and HWT has been used in

paleoecological reconstructions of surface wetness (e.g., Loisel et al. 2009). In our dataset the strength of the relationship was weaker than previously reported. For instance, Loader et al. (2016) reported $R^2 = 54\%$ for *S. magellanicum* in a single site. Given the characteristics of our data (large-scale, circumpolar), the explanatory power ($R^2_{marginal} = 17\%$) can be considered acceptable and comparable to other proxies such as testate amoebae (16% in Loader et al. 2016; Sullivan and Booth 2011). Our results imply that isotopic signals of peatland wetness in hummock-dwelling species (such as *S. fuscum*

coll.) may be weaker, or absent, compared to lawn species. It is therefore important that the same species, or species type (e.g., lawn species as they likely have a broad HWT niche), are used if $\delta^{13}C$ values are employed as a proxy to infer changes in HWT.

We also identified evidence that evapotranspiration (ET), and productivity (NPP) modify $\delta^{13}C$ values. ET and temperature

control $\delta^{13}C$ by increasing water loss at the moss surface and reducing the diffusive resistance (i.e., reducing $CO_2$ limitation), which enables discrimination against $^{13}C$ (Williams and Flanagan 1996). This mechanism requires the moss surface to partially dry out under high evaporative demand, which only occurs in hollow-lawn species and not, or to a much lesser extent, in hummock species due to high water retention, strong capillarity forces, and reduced boundary layer conductance. This can explain the stronger effect of ET/P and temperature (i.e. net water loss) on $\delta^{13}C$ in *S. magellanicum*i. NPP only

explained a small proportion of the variation in $\delta^{13}C$ values but the relationship was apparent across species. Several studies have proposed the use of $\delta^{13}C$ relative abundance to infer *Sphagnum* productivity (e.g., Rice and Giles 1996, Rice 2000, Munir et al. 2017) and our study is the first to test this at the pan-Holarctic scale. Deane-Coe et al. (2015) investigated $\delta^{13}C$ values across moss species (including *Sphagnum*) and years at one site and found a weak relationship between productivity and $\delta^{13}C$ values ($R^2=0.10$ and $0.31$, respectively). Similarly, Rice (2000) reported that relative growth rate explained about

25% of the variation in $^{13}C$ discrimination. We did not find as strong a relationship ($R^2 <0.12$), but our study was less controlled and thereby influenced by many unknown factors. Nevertheless, our results indicate independent effects of evaporation and productivity on $\delta^{13}C$ values. The lack of a strong NPP pattern somewhat limits the ability to infer productivity of *Sphagnum* in paleoecological studies.



### 4.2 Global patterns of $\delta^{18}O$ values in *Sphagnum*

Modelled $\delta^{18}O$ values in precipitation (Bowen 2010) explained much of the variation in $\delta^{18}O_{tissue}$ values among sites ($R^2$=68% for annual mean $\delta^{18}O_{precip}$). The percent variance explained was even higher if the spring period for modelled $^{18}O_{precip}$ was used, but lower for the growing season average. This result does not necessarily mean that spring season water was utilised by the plants during the growing season. Between-site variation in $^{18}O_{precip}$ values are much larger in the winter (Figure S1, see end of document), more effectively discriminating maritime and continental regions  (Bowen 2010). The better fit may

simply be an effect of a more distinct separation of $^{18}O_{precip}$ in the winter data. In contrast, the data did not support a negative correlation between precipitation and $\delta^{18}O_{tissue}$ values, and $\delta^{18}O_{tissue}$ values were weakly affected by predictors associated with water loss (ET/P and/or temperature) and species identity. The indication of $^{18}O$ enrichment in *S. magellanicum* due to ET/P was expected as the lighter isotope $^{16}O$ needs less energy to vaporize. However, the opposite trend was suggested for *S. fuscum* and surprisingly, higher surface temperatures decreased $^{18}O$ enrichment. Hence we conclude that climatic variables

associated with water loss were weak predictors after controlling for $\delta^{18}O_{precip}$ values. This result may not be too unexpected as lab experiments have so far failed to relate $^{18}O$ enrichment in *Sphagnum* to differences in evaporation rates (Brader et al. 2010).

There have been few regional studies on moss $\delta^{18}O_{tissue}$ values that span gradients of $\delta^{18}O_{precip}$ values (Royles et al. 2016,

Skrzypek et al. 2010) and most interpretations of moss $\delta^{18}O_{tissue}$ - climate relationships come from peat core studies (e.g. van der Knaap et al. 2011). In antarctic non-*Sphagnum* peat banks variation in $\delta^{18}O_{cellulose}$ values tracked $\delta^{18}O$ values in moss water across a latitudinal gradient (61°S-65°S) despite a lack of difference in $\delta^{18}O_{precip}$. This result led Royles et al. (2016) to suggest that moss water and tissue $\delta^{18}O$ values are better temporal integrators of source water than point rainfall measurements. The authors interpreted site-to-site differences as relating to differential evaporative enrichment and other

physio-chemical factors that affect $^{18}O$ exchange, fixation and biochemical synthesis. Similar patterns may also occur along elevational gradients as $\delta^{18}O_{tissue}$ values are consistent with expected isotopically heavier source water at high elevations controlling tissue signals, but with small sample sizes (n=7) patterns remain unclear (Skrzypek et al. 2010). The present study provides a much greater range of geographical and environmental variation, and arrives at similar conclusions — $\delta^{18}O_{tissue}$ values in *Sphagnum* strongly track source water.


Interestingly, the relationship between $\delta^{18}O_{tissue}$ and $\delta^{18}O_{precip}$  values detected here is very similar to that proposed some time ago by Epstein et al. (1977); $\delta^{18}O_{cellulose} = 27.33 + 0.33 \times \delta^{18}O_{precip}$ [note that Jones et al. 2014 show high correspondence between $\delta^{18}O_{cellulose}$ and $\delta^{18}O_{tissue}$ values]). However, our data suggest a slightly steeper slope and lower intercept, particularly for *S. magellanicum*. The species effect on $\delta^{18}O$ suggests a difference in the degree of evaporation from the plant surface

prior to uptake of water. The lower $\delta^{18}O$ values for *S. magellanicum* compared to *S. fuscum* (-0.83‰), is comparable to the results from bogs in Canada for the same species (-2.2‰, Aravena and Warner 1992) and between a hollow and a hummock





species in The Netherlands (-2‰, Brenninkmeuer et al. 1982). This suggests that the absorbed water in this *S. magellanicum* was subject to less evaporation. *Sphagnum* plants control the water available on their surface largely by capillarity, water storage and reducing conductance with compact morphology. Plant traits that enhance these functions are more pronounced

in species and individuals found at high HWT as these characteristics maintain high tissue water content (Hayward and Clymo 1982, Laing et al. 2014, Waddington et al. 2015). Consequently, during droughts, *Sphagnum* species growing close to the water table will dry out quickly as the evaporative demand cannot be balanced, and simultaneously photosynthesis is shut down. *Sphagnum* species higher above the water table wick water from below and store water effectively, thereby remaining photosynthetically active while water is lost due to evaporation. This mechanism would result in $^{18}$O enrichment being

higher above the water table (Brenninkmeuer 1982, Aravena and Warner 1992), and explains the positive relationship between HWT and $\delta^{18}$O in *S. magellanicum* reported by Loader et al. (2016) along a 10 m transect. We found a weak positive relationship of $\delta^{18}$O with HWT, which suggests that HWT cannot entirely explain species-specific differences in $^{18}$O enrichment. Instead, this can be attributed to lower water retention (i.e. higher evaporation at the same water deficit) in *S. magellanicum* compared to *S. fuscum* (Clymo 1973, McCarter and Price 2014). Although species differences in $^{18}$O have

been reported (Aravena and Warner 1992, Zanazzi and Mora 2005, Bilali et al. 2013), our study suggests that the species-specific $\delta^{18}$O signals may not simply be a consequence of growing at different HWT but can rather reflect distinct water retention capacity in these species.

The strong influence of $\delta^{18}$O$_{precip}$ values and, to a much lesser extent, environmental variables related to water loss, combined

with a relatively small within-site variation in $\delta^{18}$O$_{tissue}$ values, suggest that macroclimatic drivers, such as precipitation inputs, largely determine the $\delta^{18}$O value of moss tissue. These results are promising for the use of oxygen isotopes in large-scale paleoecological reconstructions (Ellis and Rochefort 2006, Chambers et al. 2012, Daley et al. 2010), although a better understanding of O isotope fractionation within tissue components and their decay relationships would improve their utility. Moreover, the simple relationships presented here can potentially be utilised to trace changes in

$\delta^{18}$O$_{precip}$ values that mirrors climate variability.

## 5 Conclusions

Our study provides new insights into large-scale variation in *Sphagnum* tissue isotopic signature and suggests that isotopic composition can be used for climatic reconstructions. We show a close link between precipitation and tissue $\delta^{18}$O values and conclude that variation in $\delta^{18}$O values are mainly driven by the macroclimate, but species differences exist. In contrast, $\delta^{13}$C

values were strongly related to local microtopography but also weakly related to macroclimate. As suggested in earlier studies, $\delta^{13}$C values were also weakly associated with NPP. These conclusions were most strongly supported for the cosmopolitan *S. magellanicum* complex and species identity should be accounted for in future C isotope studies to avoid spurious conclusions.





## 6 Data and code availability

Data and R-script to reproduce results are available for review at https://github.com/ggranath/isotopeSphagnum . Upon acceptance these files will be moved to a permanent public repository.

## 7 Supplementary material

Figure S1 and Table S1.

## 8 Author contribution

SKR, GG, HR initiated the study and formulated the research objectives. All authors were involved in data collection and SKR, NB, KP, AV, DPG performed the isotope analyses. GG performed the statistical analyses and wrote the first draft with input from SKR and HR. All authors read and commented on the manuscript and approved the final version.

## 9 Competing interests

The authors declare that they have no conflict of interest.

## 10 Acknowledgement


To the memory of coauthor and *Sphagnum* enthusiast Maria Noskova, who passed away, tragically, before this paper was finished. We thank the U.S. National Science Foundation for providing funding for Union isotope ratio mass spectrometer and peripherals (NSF-MRI #1229258) and Sarah Katz for laboratory assistance. Data collection was supported by the Russian Science Foundation, grant 14-14-00891 and by the Russian Foundation for Basic Research according to the research
projects No 14-05-00775 and No 15-44-00091, University of Ferrara (FAR 2013 and 2014), the Polish National Centre for Research and Development within the Polish-Norwegian Research Programme within the project WETMAN (Central European Wetland Ecosystem Feedbacks to Changing Climate Field Scale Manipulation, Project ID: 203258), institutional research funds from the Estonian Ministry of Education and Research (grant IUT34-7), the Natural Sciences and Engineering Research Council of Canada Discovery Grants program awarded to JLB, an NSERC Strategic Grant, and with
generous support awarded to L.I.H from the W. Garfield Weston Foundation Fellowship for Northern Conservation, administered by Wildlife Conservation Society (WCS) Canada, and National Science Foundation (NSF-1312402) to SMN. We acknowledge the Adirondack and Maine offices of The Nature Conservancy, the Autonomous Province of Bolzano (Italy), Staatsbosbeheer and Landschap Overijssel (the Netherlands), the Greenwoods Conservancy, NY and the University of Orono, ME for access to field sites.



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

| | Species | $N_{site}$ | $N_{obs}$ | Within-site | | Between-site | |
|---|---|---|---|---|---|---|---|
| | | | | Std. Dev. | Proportion variance (%) | Std. Dev. | Proportion variance (%) |
| $\delta^{13}$C | S. fuscum | 80 | 169 | 0.9 | 56 | 0.8 | 44 |
| | S. magellanicum | 83 | 168 | 1 | 51 | 0.98 | 49 |
| | | | | | | | |
| $\delta^{18}$O | S. fuscum | 80 | 168 | 0.7 | 13 | 1.83 | 87 |
| | S. magellanicum | 83 | 167 | 0.67 | 10 | 2 | 90 |

**Table 1. Sample sizes, standard variation and overall partitioning of measured variation for each species.**

| variable | effect | F | DF | P | N | R2site | R2marginal |
|---|---|---|---|---|---|---|---|
| HWT | | 26.8 | 1, 67 | <0.001 | 311 | | |
| species | -0.88+-0.13* | 43 | 1, 274 | <0.001 | | | |
| HWT*species | S.fus: -0.021+-0.008 S.mag: -0.045+-0.008 | 6 | 1, 235 | 0.01 | | 0.32 | 0.18 |
| HWT [S.magellanicum] | -0.04+-0.008 | 26 | 1, 134 | <0.001 | 158 | 0.33 | 0.17 |
| NPP | 0.0023+-0.0005 | 23.1 | 1, 309 | <0.001 | 318 | 0.12 | 0.07 |
| species | -0.38+-0.12 | 9.4 | 1, 266 | <0.01 | | 0.01 | 0.01 |
| NPP*species | | 0.7 | 1, 290 | 0.41 | | | |
| HWT | | 25.5 | 1, 246 | <0.001 | 295 | | |
| species | -0.62+-0.13* | 22.1 | 1, 269 | <0.001 | | | |
| NPP | 0.0022+-0.0005 | 22.4 | 1, 281 | <0.001 | | 0.11 | 0.07 |
| HWT*species | S.fus: -0.012+-0.007 S.mag: -0.042+-0.007 | 10.6 | 1, 267 | <0.01 | | 0.2 | 0.12 |
| elevation | 0.00035+-0.0002 | 2.7 | 1, 81 | 0.11 | 295 | 0.03 | 0.01 |
| elevation*species | | 0.5 | 1, 233 | 0.47 | | | |
| ET/P | | 0.2 | 1, 90 | 0.66 | 295 | | |
| (ET/P)*species | S.fus: -0.33+-0.40 S.mag: 0.78+-0.44 | 5 | 1, 266 | 0.03 | | -0.02 | 0.01 |
| P | -0.00013+-0.0006 | 0 | 1, 80 | 0.83 | 295 | -0.01 | 0 |
| P*species | | 1.2 | 1, 248 | 0.27 | | | |
| T | | 0 | 1, 91 | 0.97 | 295 | | |
| T*species | S.fus: -0.051+-0.034 S.mag: 0.087+-0.041 | 9.7 | 1, 273 | <0.01 | | -0.05 | 0.02 |

*The effect of S. magellanicum compared to S.fuscum at HWT 28 cm.

**Table 2. Results from linear mixed-models for $\delta^{13}C$ values. Statistical tests are based on type-2 *F*-test using Kenward-Roger adjusted degrees of freedom. The second model only included *S. magellanicum*. Elevation [m asl] and the three climatic variables (growing season sums and means: ET/P, P [mm], temp [°C]) were tested one by one in the model including HWT [Height above the Water Table, cm], species and NPP [mg m$^{-2}$ year$^{-1}$]. Estimated effects (+/- SEs) are only given for main effects if interactions were considered negligible. These effects are slopes for continuous variables (all variables except species) and for species (categorical) the difference between *S. magellanicum* and *S. fuscum* (i.e. *S. fuscum* being the reference level). In the presence of an interaction between HWT and species, the species effect was estimated at mean HWT. $R^2_{site}$ = explained between site variance, $R^2_{marginal}$ = explained total variance.**





| variable | effect | F | DF | P | N | R2site | R2marginal |
|---|---|---|---|---|---|---|---|
| annual precip d18O | 0.43+-0.035 | 148.4 | 1, 95 | <0.001 | 335 | 0.69 | 0.5 |
| species | -0.83+-0.083 | 101.3 | 1, 250 | <0.001 | | | 0.05 |
| annual precip d18O*species | | 1.9 | 1, 261 | 0.16 | | | |
| Apr-Sep precip d18O | 0.49+-0.049 | 100.5 | 1, 94 | <0.001 | 335 | 0.58 | 0.42 |
| species | -0.83+-0.083 | 99.4 | 1, 249 | <0.001 | | | 0.05 |
| Apr-Sep precip d18O*species | | 1.4 | 1, 256 | 0.24 | | | |
| Jan-Apr precip d18O | 0.37+-0.027 | 187.2 | 1, 96 | <0.001 | 335 | 0.75 | 0.55 |
| species | -0.84+-0.083 | 102.3 | 1, 252 | <0.001 | | | 0.05 |
| Jan-Apr precip d18O*species | | 2.3 | 1, 265 | 0.13 | | | |
| annual precip d18O | 0.41+-0.038 | 111.9 | 1, 88 | <0.001 | 310 | 0.64 | 0.46 |
| HWT | 0.015+-0.005 | 10.4 | 1, 288 | <0.01 | | 0 | 0.01 |
| ET/P | | 0.1 | 1, 99 | 0.81 | 335 | | |
| (ET/P)*species | S.fus: -0.39+-0.48 S.mag: 0.28+-0.50 | 3.5 | 1, 266 | 0.06 | | 0 | 0 |
| P | -0.0005+-0.0008 | 0.4 | 1, 99 | 0.54 | 335 | 0 | 0 |
| P*species | | 0.8 | 1, 257 | 0.37 | | | |
| T | -0.14+-0.051 | 7.4 | 1, 96 | 0.01 | 335 | 0.02 | 0.02 |
| T*species | | 1.6 | 1, 274 | 0.21 | | | |

**Table 3. Results from linear mixed-models for δ¹⁸O values. Statistical tests are based on type-2 *F*-test using Kenward-Roger
adjusted degrees of freedom. Three time periods for modelled δ¹⁸O values (‰) in precipitation were tested individually: annual
mean, growing season (Apr-Sep) and spring (Jan-Apr). The three climatic variables (growing season sums and mean: ET/P, P
[mm], temp [°C]) were tested one by one in a model including HWT [cm] and mean annual δ¹⁸O values). Estimated effects (+/-
SEs) are only given for main effects if interactions were considered of negligible. These effects are slopes for continuous variables
(all variables except species) and for species (categorical) the difference between *S. magellanicum* and *S. fuscum* (i.e. *S. fuscum*
being the reference level). R²<sub>site</sub> = explained between site variance, R²<sub>marginal</sub> = explained total variance.**



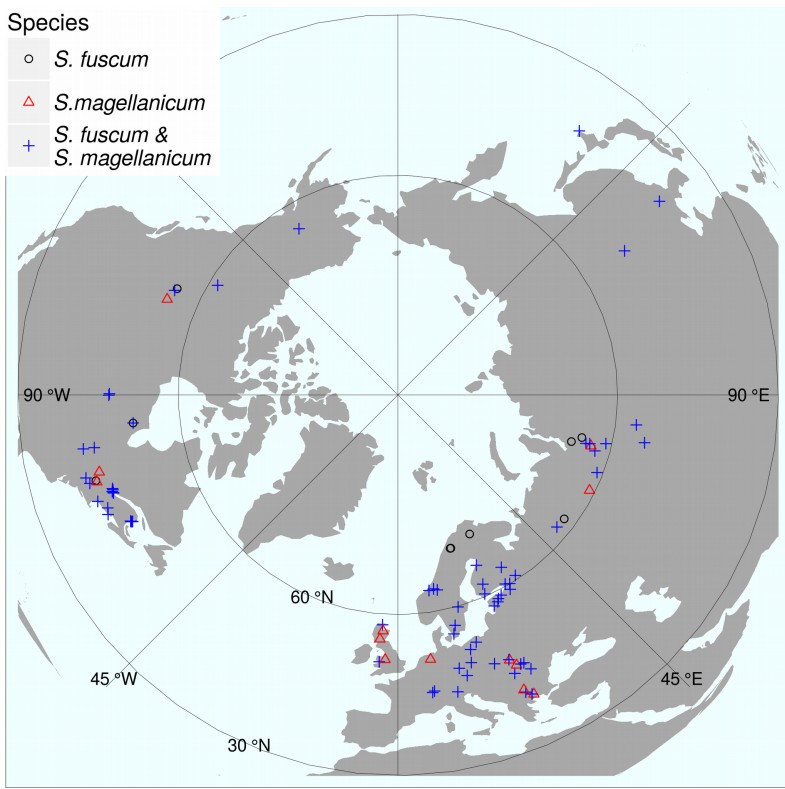

**Figure 1. Map illustrating sample sites for the investigated species. At some sites only one of the two *Sphagnum* species was sampled, indicated by red triangles or black circles, otherwise sites contained both species (blue crosses). The map is centered on the North Pole and has an orthographic projection. Geographical ranges: latitude 41.6N-69.1N, elevation 2 - 1 829 m asl. See Table S1 for details.**






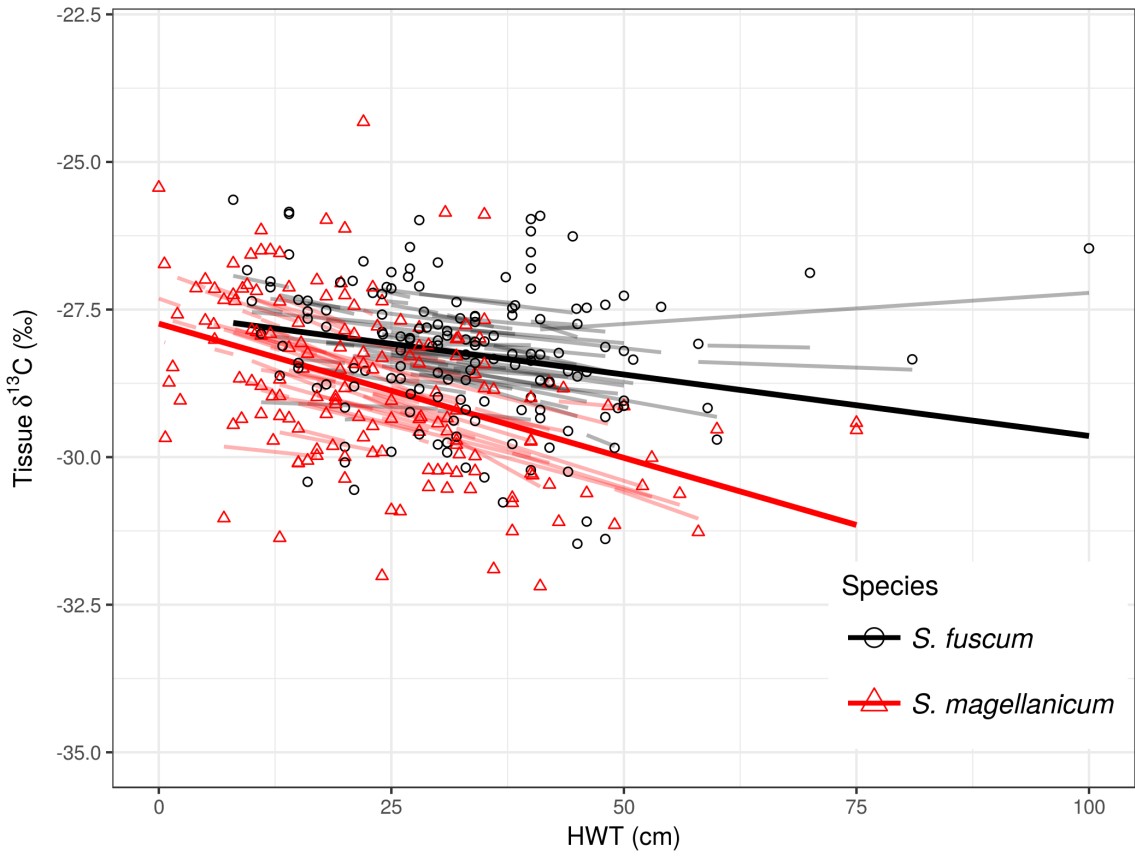

**Figure 2. Relationship between height above the water table (HWT, measured at the end of the growing season) and δ$^{13}$C values in two *Sphagnum* species sampled across the Holarctic realm. Pale lines represent relationships for individual sites, while thicker lines show the pooled regression line in a mixed-effect model. Equations: *S. fuscum*, δ$^{13}$C = -27.56 - 0.021 × HWT; *S. magellanicum*, δ$^{13}$C = -27.74 - 0.045 × HWT. N$_{site}$ = 83, N$_{total}$ = 311. See also Table 1.**





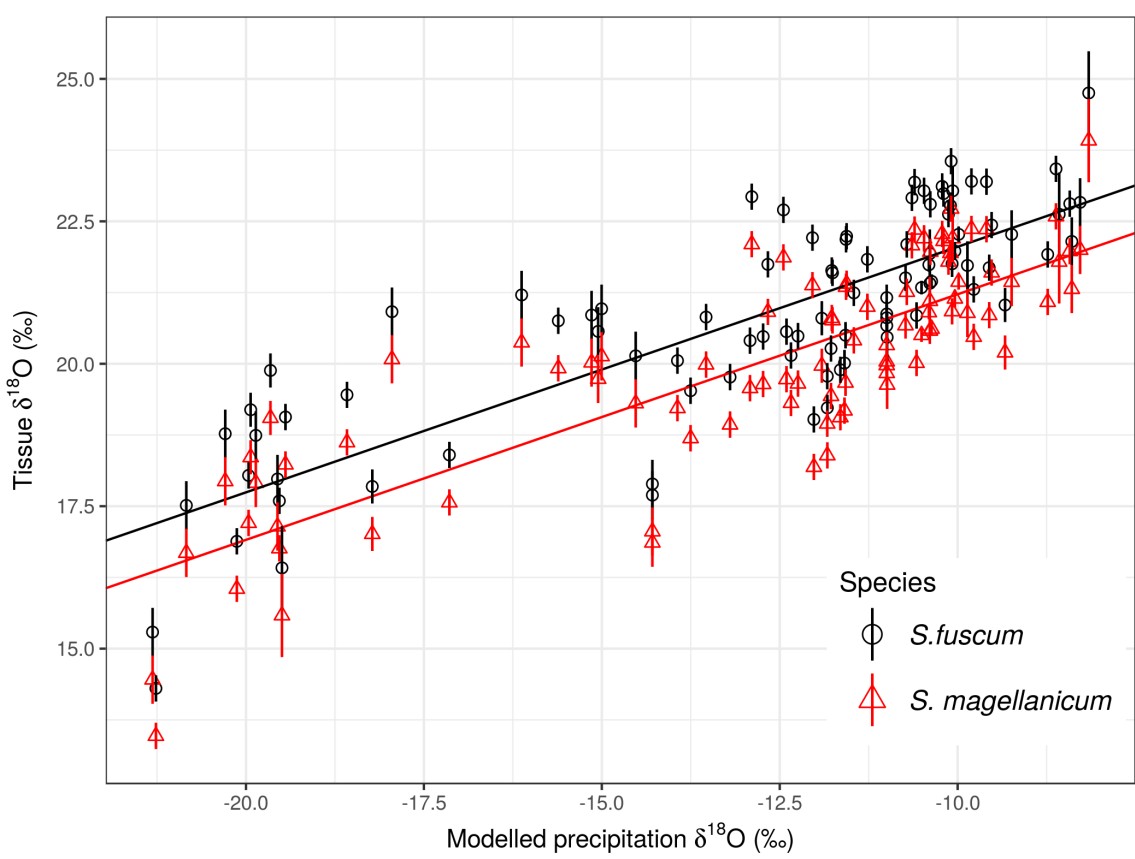

**Figure 3. Association between modelled annual mean $\delta^{18}O_{precip}$ values and $\delta^{18}O$ values in two *Sphagnum* species. Data show predicted site means (BLUPs) and error bars represent the approximate 95% confidence intervals (2 x SE). Regression lines with different intercepts (P <0.001, Table 2) illustrate the relationship between modeled $\delta^{18}O_{precip}$ and *Sphagnum* $\delta^{18}O$. Equations: *S. fuscum*, 26.36 + 0.43 × $\delta^{18}O_{precip}$ (n= 1-2, $N_{site}$ = 80); *S. magellanicum*, 25.53 + 0.43 × $\delta^{18}O_{precip}$ (n= 1-2, $N_{site}$ = 83).**