# Peer review of "Environmental and taxonomic controls of carbon and oxygen stable isotope composition in *Sphagnum* across broad climatic and geographic ranges"

_Biogeosciences, 2018_

## Short Comment (SC1) · 20 Apr 2018

General comments: The manuscript by Granath et al presents a large northern dataset of d13C and d18O from Sphagnum magellanicum and Sphagnum fuscum tissues. Results show promise for d18O from plant tissue as a proxy for d18O from precipitation; the relationship between tissue and source water could be used broadly to reconstruct changes in precipitation from peat core records. The relationship between d13C from plant tissue and environmental conditions was shown to be a little more complicated to interpret because of species-specific differences and confounding factors (water table

and NPP primarily). The dataset presented here is coherent and spans a broad range of climatic conditions. To my knowledge, the statistical tests performed are adequate and provide honest/reliable results. In general, this is a much needed review of what is known (and what remains unclear) about the relationships between environmental conditions and the stable isotope signature of Sphagnum tissues. This synthesis might help us better understand Sphagnum physiology and its adaptation to local conditions. Also, the text reads well and should be well received by the BG audience and particularly by the terrestrial ecosystem ecology and paleoclimatology communities. I recommend publication of this manuscript pending that the following specific and technical comments be considered in the final article:

Specific comments: (1) the use of Sphagnum CAPITULUM for the analysis – we know that many other authors have used stems OR leaves in the past and that these 2 types of tissues have different d13C values (see work by Loader for a discussion on the offset); also, there might be translocation from the apex down to the stems and leaves (see work by Bragazza) – I wonder what a difference it makes to analyze the capitulum vs. the top part of the stem. Could this partly explain the relatively wide spread of data you obtained with d13C?

(2) water table measurements at the END of the growing season: I wonder if part of the somewhat weak relationship between d13C and HWT could be caused by a contracted spread of HWT? Assuming that the dry end of the microhabitat remains dry throughout the growing season, but that the wetter microhabitats tend to dry out over time, it is possible that measuring HWT at the end of summer does not provide an accurate picture. If photosynthesis was to preferentially occur early during the growing season (i.e., under wetter conditions) and then stop, the pattern you observe could be in part explained by a sampling bias.

(3) I'd like a precision on the bulk density measurements: it is said that the top 30mm of the stem were used to calculate BD; how was volume determined?

(4) Figure 2: A discussion on regional differences that you found across your dataset would be useful. For example, are there areas where d13C and HWT was more strongly correlated then others? what about d13C and NPP? Or was NPP more strongly correlated for wetter samples? Same goes with d18O and P, as well as d18O and Evaporation: is there anything else that could be learned from within your dataset?

(5) Figure 3: i'm curious to know more about the d18O values between -20 and -15 permille; they are almost all poorly predicted by your linear model. Where do they come from? what might explain their 'unusual' signature?

Technical corrections:

line 204: add a space between and_are

line 251: we first builT (change the builD for builT) ... and WERE identified (change ARE for WERE). Everything else in here uses past tense

line 286-287: "S. magellanicum..." should follow the previous sentence; there is currently a 'line jump'.

Julie Loisel April 20 2018

―――――――――――――――――――――――

---

## Referee Comment (RC2) · J. Loiesel (Referee) · 28 May 2018

See my short comment from 20 Apr 2018.
* * *

---

## Referee Comment (RC3) · Anonymous Referee #3 · 6 Jun 2018

This paper addresses important questions about how the isotope composition of Sphagnum is controlled by environmental conditions: which is key to using peat bogs as palaeoenvironmental archives. The authors have chosen two cosmopolitan species, which allows the important subject of species specificity of signals to be addressed, and have a good distribution of sample locations from around the Holarctic, the regions where using Sphagnum as a paleoclimate archive is potentially feasible. The differential sensitivity of the two Sphagnum species to the environmental variables is an important result, even when growing in close proximity. The relationship between

18O and the annual precipitation is interesting and an important result that has wide ranging relevance. The relationships with 13C are much more complicated! I think that to make the most of the data set, the results section needs to be expanded, with more description of the raw data, which will give the reader a better understanding of the data. The impact of the many environmental variables is very complex, hence several of the relationships have very low explanatory power: there needs to be more critical analysis of the statistics teasing out those that have clear biological relevance. One aspect on which there is no comment is any regional variation in values / relationships, which would be interesting. Specific comments:

Line 72: I think it would be better to replace "elements" with "compounds", as although it is isotopic composition of C and O being analysed, they are not abstracted from the atmosphere in their elemental form, and are analysed within compounds.

Line 73: compositions (rather than composition)

Line 74: Can be difficult to determine it the material is dead, some may spontaneously start to regrow if exposed to light.

Line 83: Holarctic spelling

Line 83: Were the differences significant in d13C between species

Line 85: Where R2 is only 6%, I'm doubtful of its importance as a significant predictor: I think this needs expansion and may be easier to leave out of the abstract

Line 90: Expand HWT and NPP at first use Introduction

Line 97: forcings (rather than forcing), responses (rather than response)

Line 101: replace "is" with "are"

Line 106-113: This paragraph is a bit unclear. It is the isotopic composition of the Co2 that is in the chloroplast, rather than purely its concentration that is important for the extent of carbon isotope composition. Thus, if the rate of diffusion is slow,

and assimilation continues by the moss, the carbon concentration will decrease, but what is more important is that the proportion of $13CO_2$ will increase and consequently discrimination against $13CO_2$ will decrease.

Line 113: remove "consequently".

Line 113: Respiratory $CO_2$ can be fixed when the mosses are not submerged: particularly close to the ground the isotopic composition of the source $CO_2$ may vary in space and time depending on the extent of mixing between any respired $CO_2$ at the bog surface, and the well mixed atmosphere above.

Line 147: $CO_2$: subscript rather than superscript

Line 155: compositions (rather than composition) Methods

Line 187: How was the end of the growing season identified?

Table S!: mark which / both species were collected from each site

Line 204: space between and and are

Line 215: However, cellulose may be more applicable for a comparison to palao studies, in which case the differential breakdown of different components means analysing a single component can incease the accuracy. Furthermore, may be a significant contributor to species specific differences. Furthermore, whilst trying to pin down influencing factors which previous studies have shown to be very complex, whilst there is a strong relationship between the composition of organic matter and cellulose, particularly for 18 0, 30-50% variation in celluloe-OM relationship is not explained using OM, and the mean annual modelled water leaves 24% to be explained. . ..cellulose maybe could have been measured and precip collected to facilitate explanations

Line 234: when were the HWT measurements made? Depending when most of the growth occurred, this could have a significant impact on both isotope relationships

Line 240: how long a period were the pins in place? The calculations for NPP need

more detail both for the amount of vertical growth, and the bulk density measurements as that can be very difficult to do accurately on loose sphagnum

Line 251: built rather than build Results: Need more details in the results section – the results need to be described at the beginning. What are the ranges of the raw data for the isotope values, what are the growth rates, bulk densities, water table depths etc.

Line 272: Table 1 is unclear: need means of both 13C and 18O values rather than just the variation. Add per mille sign to SD values. Unclear what the proportion of variance is referring to: is this the variance explained by the mixed effect model?

Line 268-272: Are the relationships between d13C and HWT significant?

Fig 2: How many samples per site into each line? If its only two per site (Nsite = 80, N=c. 160), is that enough info for a valid calculation: I'm not convinced the site lines are meaningful. Plot confidence interval on pooled regression lines. The individual site lines make it hard to see the overall averages.

Fig 3: Plot confidence interval on regression lines.

Discussion

Line 324: This overstates the influence that you measured on d13C, especially of ET, which had "weak evidence" for S magellenicum.

Line 335 "influenced by many unknown factors": could this be expanded and made a little more specific?

Line 345: Do you mean precipitation amount?

Line 373: Sphagnum doesn't actively control the water availability: it is a passive process, influenced by growth form etc. I think that "control" implies that it is an active process.

Line 391: May not be generalisable across moss species: sphagnums are generally

wet so tightly coupled to the source water, mosses which rapidly hydrate and desiccate repeatedly may be less tightly coupled to the source water and more dependent on evaporative processes.

---

## Author Response (AR1)

**LETTER TO THE EDITOR**

Dear Dr. van der Meer,

Thank you for your comments and for your continued interest in our manuscript. We have modified the manuscript to address the suggestions of the referees. We believe this has improved the clarity and presentation of the manuscript and we hope that these changes align with your expectations. After consideration we included the new table on summary statistics as supplemental material as much of the statistics already existed in the manuscript (mostly in figures). Statistics on height increment and bulk density have been added in the text.

As you suggest, the  $\delta^{13}$ C story is complex as evidenced by the modest relationships exhibited with environmental factors. *Sphagnum* mosses have no known CO2 concentrating or transport mechanisms. Studies of physiology (i.e., real-time 13C discrimination during photosynthesis), controlled growth under different environmental conditions, and field values at different water availabilities have confirmed the importance of extra-cellular water as a major determinant of 13C/12C fractionation. Indeed, the effect of external water was strong enough to overwhelm variation in the structure and placement of photosynthetic cells in a comparison of two *Sphagnum* species (Rice and Giles 1996, PC&E 19:118). The role of diffusion through the liquid phase and its importance as a limit to carbon uptake in bryophytes has been recently reviewed by Hanson et al. (2014, Ch. 6 in Hanson and Rice [eds.] Photosynthesis in Bryophytes and Early Land Plants, Springer).

The results of the present manuscript support that this process remains important when considering even broad geographic regions and within two species. However, there remains significant variation in  $\delta^{13}$ C unexplained by either height above water table or productivity. In section 4.1, we discuss underlying causes that may contribute to that variation. It should be noted that the two species studied were rarely found submerged and differences are not likely due to what is dissolved within pore water in peat—either respired

 $CO_2$  or other organic compounds that may be taken up. However, respired carbon in peatlands may alter both the isotopic composition and the concentration of  $CO_2$  in the atmosphere near the surface where it can be refixed (see Turetsky and Weider 1999, Ecoscience 6:587, a reference we have added). We have clarified this in the first paragraph of section 4.1 where we discuss it. Although this might contribute to the variation between sites, it is unlikely that this caused the within site variation, which was higher for both species. Unfortunately, we are not aware of studies that have explored within site variation in refixed  $CO_2$  to allow us to elaborate on its possible effects.

We appreciate your input and welcome any additional suggestions you may have. Thank you for your time.

Gustaf Granath and co-authors

**RESPONSE TO REVIEW COMMENTS**

These are in general the responses posted online, but with a few clarifications on the made changes.

**REVIEWER #1**

J. Loisel

General comments: The manuscript by Granath et al presents a large northern dataset of d13C and d18O from Sphagnum magellanicum and Sphagnum fuscum tissues. Results show promise for d18O from plant tissue as a proxy for d18O from precipitation; the relationship between tissue and source water could be used broadly to reconstruct changes in precipitation from peat core records. The relationship between d13C from plant tissue and environmental conditions was shown to be a little more complicated to interpret because of species-specific differences and confounding factors (water table and NPP primarily). The dataset presented here is coherent and spans a broad range of climatic conditions. To my knowledge, the statistical tests performed are adequate and provide honest/reliable results. In general, this is a much needed review of what is known (and what remains unclear) about the relationships between environmental conditions and the stable isotope signature of Sphagnum tissues. This synthesis might help us better understand Sphagnum physiology and its adaptation to local conditions. Also, the text reads well and should be well received by the BG audience and particularly by the terrestrial ecosystem ecology and paleoclimatology communities. I recommend publication of this manuscript pending that the following specific and technical comments be considered in the final article:

Specific comments:

(1) the use of Sphagnum CAPITULUM for the analysis – we know that many other authors have used stems OR leaves in the past and that these 2 types of tissues have different d13C values (see work by Loader for a discussion on the offset); also, there might be translocation from the apex down to the stems and leaves (see work by Bragazza) – I wonder what a difference it makes to analyze the capitulum vs. the top part of the stem. Could this partly explain the relatively wide spread of data you obtained with d13C?

RESPONSE: We used branches in the capitula for our comparisons because we were interested in matching isotope values with environmental conditions in the present growing season. Given our broad sampling including sites that could experience slow growth due to cold temperatures or water stress, we opted to measure tissue that most likely would reflect recent conditions. Loader et al. (2016; J Quat Res 31:426) also used capitula in a study comparing microclimates and Sphagnum isotope values, presumably for the same reason although not explicitly stated.We prepared samples from 10 capitula from each plot. Loader et al. (2016; above) found a 1.7 per mil range in d13C among 102 Sphagnum capitula growing within a 20 cm2 area. We believe our sampling reflects this naturally occurring variation and contributed to the spread of d13C values.

Loader et al. (2007; The Holocene 17:403) show that the carbon isotopic values in branch types (hanging vs pendant branches) differ consistently, but that the difference is small (0.26 per mil). However, there is a much greater difference between these and stems (>1 per mil). Moschen et al. (2009; Chemical Geology 259:262) found a similar offset.

In summary, for absolute isotopic values and its variation it matters what part of the plant is analysed. However, relationships should remain the same. We added some text to inform the reader that they should be aware of this when interpreting/applying these relationships.

(2) water table measurements at the END of the growing season: I wonder if part of the somewhat weak relationship between d13C and HWT could be caused by a contracted spread of HWT? Assuming that the dry end of the microhabitat remains dry throughout the growing season, but that the wetter microhabitats tend to dry out over time, it is possible that measuring HWT at the end of summer does not provide an accurate picture. If photosynthesis was to preferentially occur early during the growing season (i.e., under wetter conditions) and then stop, the pattern you observe could be in part explained by a sampling bias.

RESPONSE: Our measurements of HWT is a snapshot of the d13C-HWT relationship and may indeed have been tighter if we had measured continuous HWT. This shortcoming is highlighted in the manuscript (second sentence section 4.1). Unfortunately, collection of continuous HWT data was not logistically possible but we argue that HWT in the end of the season is a good proxy for relative HWT differences among locations. Growth mainly occurs in late summer/fall in temperate and boreal regions and therefore HWT at the the end of the season is assumed to be a better proxy of relative HWT during growth than spring HWT. We did measure HWT in the spring as well and found spring HWT and fall HWT to be strongly correlated (r=0.74, this number has now been added).

(3) I'd like a precision on the bulk density measurements: it is said that the top 30mm of the stem were used to calculate BD; how was volume determined? RESPONSE: Also requested by reviewer #3. A known area was cut out carefully to avoid compaction (diameter=10cm). Stems within this area were then trimmed to 30 mm by cutting off the capitula and the lower part. This has now been clarified in the method section.

(4) Figure 2: A discussion on regional differences that you found across your dataset would be useful. For example, are there areas where d13C and HWT was more strongly correlated then others? what about d13C and NPP? Or was NPP more strongly correlated for wetter samples? Same goes with d18O and P, as well as d18O and Evaporation: is there anything else that could be learned from within your dataset?

RESPONSE: These are all good suggestions for further exploratory analyses and also put forward by reviewer #3. It is, however, important to point out that such analyses are of exploratory character as we may find spurious relationships when we perform subset analyses with small sample sizes. The NPP x HWT interaction was actually tested but we missed to give this result (in the text it says "we removed negligible interactions"). This has been fixed (see Table heading). In the revised version we discuss areas/data points that do not fit the overall patterns.

(5) Figure 3: i'm curious to know more about the d18O values between -20 and -15 permille; they are almost all poorly predicted by your linear model. Where do they come from? what might explain their 'unusual' signature? *RESPONSE: Samples with these values occur in continental interiors both in Canada (Northwest Territories,NWT) and in West Siberia. As the d18O model suggests, these sites are expected to experience precipitation with low d18O values. In fact, the linear relationship looks pretty robust along the d18O values. One value, from NWT, does have a much lower value than predicted but we have no idea why except that the 18Oprecip model is less accurate in this region. We added some of these details in the revised version.*

Technical corrections:

line 204: add a space between and\_are line 251: we first builT (change the builD for builT) ... and WERE identified (change ARE for WERE). Everything else in here uses past tense line 286-287: "S. magellanicum..." should follow the previous sentence; there is currently a 'line jump'.

RESPONSE: Thank you for pointing out these correction.

**REVIEWER #3**

Anonymous

This paper addresses important questions about how the isotope composition of Sphagnum is controlled by environmental conditions: which is key to using peat bogs as palaeoenvironmental archives. The authors have chosen two cosmopolitan species, which allows the important subject of species specificity of signals to be addressed, and have a good distribution of sample locations from around the Holarctic, the regions where using Sphagnum as a paleoclimate archive is potentially feasible. The differential sensitivity of the two Sphagnum species to the environmental variables is an important result, even when growing in close proximity. The relationship 18O and the annual precipitation is interesting and an important result that has wide ranging relevance. The relationships with 13C are much more complicated! I think that to make the most of the data set, the results section needs to be expanded, with more description of the raw data, which will give the reader a better understanding of the data. The impact of the many environmental variables is very complex, hence several of the relationships have very low explanatory power: there needs to be more critical analysis of the statistics teasing out those that have clear biological relevance. One aspect on which there is no comment is any regional variation in values / relationships, which would be interesting.

RESPONSE: Also reviewer #1 pointed out the need for a more detailed result section and further analyses on regional differences. We agree that this is a useful addition and the revised version we include an overview of the variables (means,SDs, ranges) and comments on data points that diverge from the overall trends (site location). Summary statistics were added as a

supplementary table (Table S2) and in the text for height increment and bulk density.

Regarding the statistical analyses: We think the reviewer refer to the d13C results, and the effect of NPP, ET and temperature that are discussed although their R2-values are rather low. We believe that NPP result is still relevant as this is an expected relationship with clear theoretical basis. Our discussion regarding ET and temperature are, however, less relevant as the explanatory power was very low and the underlying mechanism not as clear. Thus, in the revised version some parts have been deleted and we only briefly discuss these variables.

**Specific comments:**

Line 72: I think it would be better to replace "elements" with "compounds", as although it is isotopic composition of C and O being analysed, they are not abstracted from the atmosphere in their elemental form, and are analysed within compounds.

RESPONSE: This sentence is changed to "..... depend on nutrients, water and CO2 uptake from the atmosphere."

Line 73: compositions (rather than composition) *RESPONSE: OK.*

Line 74: Can be difficult to determine it the material is dead, some may spontaneously start to regrow if exposed to light.

RESPONSE: Correct. 'dead' has been removed from the sentence.

Line 83: Holarctic spelling *RESPONSE: Will be fixed.*

Line 83: Were the differences significant in d13C between species *RESPONSE: Yes, but we prefer to avoid P-values in the abstract.*

Line 85: Where R2 is only 6%, I'm doubtful of its importance as a significant predictor: I think this needs expansion and may be easier to leave out of the abstract

RESPONSE: As the relationship between d13C and NPP was a part of our aims, we would like to include this result in the abstract. We added a few words about the poor strength of this relationship on L87.

Line 90: Expand HWT and NPP at first use Introduction RESPONSE: We assume the reviewer means 'Abstract' here (L90 is in the Abstract section). Regardless, HWT and NPP are written out at first use, both in the Abstract and the Introduction.

Line 97: forcings (rather than forcing), responses (rather than response) *RESPONSE: OK.*

Line 101: replace "is" with "are" *RESPONSE: OK.*

Line 106-113: This paragraph is a bit unclear. It is the isotopic composition of the Co2 that is in the chloroplast, rather than purely its concentration that is important for the extent of carbon isotope composition. Thus, if the rate of diffusion is slow, and assimilation continues by the moss, the carbon concentration will decrease, but what is more important is that the proportion of 13CO2 will increase and consequently discrimination against 13CO2 will decrease.

RESPONSE: Yes, this is what we meant and it is mentioned in the following sentences. It has been clarified in the revised version.

Line 113: remove "consequently". *RESPONSE: We kept this.*

Line 113: Respiratory CO2 can be fixed when the mosses are not submerged: particularly close to the ground the isotopic composition of the source CO2

may vary in space and time depending on the extent of mixing between any respired CO2 at the bog surface, and the well mixed atmosphere above. *RESPONSE: This is correct. This potential mechanism has been included in the revised version with references (eg Limpens et al. Journal of Vegetation Science 19(6):841-848. 2008, https://doi.org/10.3170/2008-8-18456)*

Line 147: CO2: subscript rather than superscript *RESPONSE: OK.*

Line 155: compositions (rather than composition) Methods *RESPONSE: OK.*

Line 187: How was the end of the growing season identified? *RESPONSE: The end of the growing season was defined as "when there is risk of snowfall or frost to occur". Of course, some sites are remotely located and it is hard for a researcher to time this. Hence, growth measurements may stop before the "true" end of the growing season. However, this last period likely has negligible growth. We describe this in the revised version.*

Table S!: mark which / both species were collected from each site *RESPONSE: An additional column indicating the species sampled at each site has been added to Table S1.*

Line 204: space between and and are *RESPONSE: OK.*

Line 215: However, cellulose may be more applicable for a comparison to palao studies, in which case the differential breakdown of different components means analysing a single component can incease the accuracy. Furthermore, may be a significant contributor to species specific differences. Furthermore, whilst trying to pin down influencing factors which previous studies have shown to be very complex, whilst there is a strong relationship between the composition of organic matter and cellulose, particularly for 18 0, 30-50% variation in celluloe-OM relationship is not explained using OM,and the mean annual modelled water leaves 24% to be explained...cellulose maybe could have been measured and precip collected to facilitate explanations

RESPONSE: We agree that cellulose extraction would have improved our ability to develop quantitative isotope-environment transfer functions that would have facilitated the connection with paleo studies. Unfortunately, this was not feasible for the present study. We believe the value of our study arises from the broad geographic sampling linking contemporary isotope signatures to environmental conditions, which is adequately addressed using isotopes derived from organic matter. In addition, given the high number of research participants, many of whom visited sites only at the start and the end of the growing season, we were unable to perform the regular rainfall collection necessary to determine annual average d180 in precipitation. Instead we relied on modelled data, which has shown to be very accurate and has the benefit that it is easy to use our results.

These arguments and explanations has been incorporated in the Method section.

Line 234: when were the HWT measurements made? Depending when most of the growth occurred, this could have a significant impact on both isotope relationships.

RESPONSE: Also commented on by reviewer #1. We here repeat the same response.

Our measurements of HWT is a snapshot and the d13C-HWT relationship may have been tighter with continuous HWT data. This is also pointed out in the manuscript (second sentence section 4.1). Now continuous HWT data was not logistically possible but we argue that HWT in the end of the season is a good proxy for relative HWT differences among locations. Growth mainly occur in late summer/fall in temperate and boreal regions and therefore HWT at the the end of the season is assumed to be a better proxy of relative HWT during growth than spring HWT. We did measure HWT in the spring as well and spring HWT and fall HWT was strongly correlated (r=0.74, this is now in the manuscript). Line 240: how long a period were the pins in place? The calculations for NPP need more detail both for the amount of vertical growth, and the bulk density measurements as that can be very difficult to do accurately on loose sphagnum

RESPONSE: Growing season (the time wires were in the field) varies among sites. Bulk density can be hard to estimate accurately but it is easier to get precise values for denser species like S.fuscum and S.magellanicum as they grow in slightly drier habitats. We will add information about mean and variation in height growth and bulk density (Result section).

Line 251: built rather than build *RESPONSE: OK.*

Results: Need more details in the results section – the results need to be described at the beginning. What are the ranges of the raw data for the isotope values, what are the growth rates, bulk densities, water table depths etc.

RESPONSE: We have added a table (Table S2) showing the means, SDs and ranges.

Line 272: Table 1 is unclear: need means of both 13C and 18O values rather than just the variation. Add per mille sign to SD values. Unclear what the proportion of variance is referring to: is this the variance explained by the mixed effect model?

RESPONSE: Table 1 shows the variation and how it is partitioned between with-site and between-site. One of the aims of the study was to investigate where most variation in isotopic variation can be found in Sphagnum. Hence, there is no need showing the means in Table 1, and similar information is given in figure 2-3. However, the table caption was brief and we added information on what it actually shows (eg the definition of proportion variance: that it is the proportion of total variance).

Line 268-272: Are the relationships between d13C and HWT significant? *RESPONSE: Yes, and this information is given in Table 2.*

Fig 2: How many samples per site into each line? If its only two per site (Nsite = 80, N=c. 160), is that enough info for a valid calculation: I'm not convinced the site lines are meaningful. Plot confidence interval on pooled regression lines. The individual site lines make it hard to see the overall averages. RESPONSE: Number samples per site varies, but is mostly two. Site is a random factor and lines show the estimated response per site. The benefit of showing individual lines is that the reader can evaluate if within-site trends follow the between site trends. Here they do so, but it does not have to be the case (think Simpson's paradox). Therefore we think it is a more honest illustration of the analyses (and data) to plot the individual lines. Confidence intervals (CIs) depict another sort of variation that can be found in Table 2 (SEs of regression coefficients). With the population level lines being close to each other, CIs for each species may be hard to distinguish for the reader. To illustrate CIs, it is probably necessary to split the figure into two panels, but then the species-specific responses may be less obvious. We agree that the average lines are hard to see because points are plotted on top of them. We have included a clearer version of this graph. See also next comment.

**Fig 3: Plot confidence interval on regression lines.**

RESPONSE: Similar to Figure 2, the two lines are rather close to each other and the confidence intervals (CIs) will be hard to distinguish. Unless we split the figure into two panels, such CIs may not be very informative for the reader. Details on the regression lines (SEs) can be found in Table 2 for the readers that want such details.

All together, we are not convinced that adding CIs will significantly improve our figures. At the same time, we don't have particularly strong opinions and if the editor prefers CIs we are open to change the figures accordingly.

**Discussion**

Line 324: This overstates the influence that you measured on d13C, especially of ET, which had "weak evidence" for S magellenicum.

RESPONSE: We agree that this was not correctly worded. The evidence for ET (and temperature) was in general weak with low R2s and we have shorten this in the new version.

Line 335 "influenced by many unknown factors": could this be expanded and made a little more specific?

RESPONSE: Good point. We have clarified this and we briefly mention the complex interactions among environmental factors that may affect Sphagnum growth across our sites.

Line 345: Do you mean precipitation amount? *RESPONSE: Yes. Now corrected.*

Line 373: Sphagnum doesn't actively control the water availability: it is a passive process, influenced by growth form etc. I think that "control" implies that it is an active process.

RESPONSE: Reworded as this is mostly a passive process.

Line 391: May not be generalisable across moss species: sphagnums are generally wet so tightly coupled to the source water, mosses which rapidly hydrate and desiccate repeatedly may be less tightly coupled to the source water and more dependent on evaporative processes. *RESPONSE: Specified that we mean peatland mosses.*

**Environmental and taxonomic controls of carbon and oxygen stable isotope composition in *Sphagnum* across broad climatic and geographic ranges**

Gustaf Granath1, Håkan Rydin1, Jennifer L. Baltzer2, Fia Bengtsson1, Nicholas Boncek3, Luca

- 5 Bragazza4,5,6, Zhao-Jun Bu7,8, Simon J. M. Caporn9, Ellen Dorrepaal10, Olga Galanina11,40, Mariusz Gałka12, Anna Ganeva13, David P. Gillikin14, Irina Goia15, Nadezhda Goncharova16, Michal Hájek17, Akira Haraguchi18, Lorna I. Harris19, Elyn Humphreys20, Martin Jiroušek21, 22, Katarzyna Kajukało12, Edgar Karofeld23, Natalia G. Koronatova24, Natalia P. Kosykh24, Mariusz Lamentowicz12, Elena Lapshina25, Juul Limpens26, Maiju Linkosalmi27, Jin-Ze Ma7,8, Marguerite Mauritz28, Tariq M. Munir29
- 30, Susan Natali31, Rayna Natcheva13, Maria Noskova†, Richard J. Payne32, 33, Kyle Pilkington3, Sean Robinson34, Bjorn J. M. Robroek35, Line Rochefort36, David Singer37,41, Hans K. Stenøien38, Eeva-Stiina Tuittila39, Kai Vellak23, Anouk Verheyden14, James Michael Waddington19, Steven K. Rice3

[revised manuscript text omitted]

Sphagnum mosses are the most dominant peat-forming plant group in acidic peatlands. The composition of stable isotopes of carbon and oxygen in their tissues is affected by different environmental conditions, operating through their impact on fractionation processes. When not submerged, carbon isotope signals in bulk tissues or components such as cellulose depend mainly on the concentration and isotopic composition of  $[CO_{24}]$  in the chloroplast  $([CO_{24}])$ , which alters isotope discrimination during biochemical fixation of CO2 and by fractionation caused by diffusion to the chloroplast (Farguhar et al. 1989, O'Leary 1988). 110 In mosses, the CO2 concentration in the chloroplast,  $[CO_2]_{c_a}$  is 
[revised manuscript text omitted]

| effect                                  | F                                                                                                                                                                                                               | DF                                                                                                                                                                                                                                                                                                                                                                                                                                                                                                                                                                                                 | Р                                                                                                                                                                                                                                                                                                                                                       | Ν                                        | R2site                                     | R2marginal                                      |
|-----------------------------------------|-----------------------------------------------------------------------------------------------------------------------------------------------------------------------------------------------------------------|----------------------------------------------------------------------------------------------------------------------------------------------------------------------------------------------------------------------------------------------------------------------------------------------------------------------------------------------------------------------------------------------------------------------------------------------------------------------------------------------------------------------------------------------------------------------------------------------------|---------------------------------------------------------------------------------------------------------------------------------------------------------------------------------------------------------------------------------------------------------------------------------------------------------------------------------------------------------|------------------------------------------|--------------------------------------------|-------------------------------------------------|
| 0.43+-0.035                             | 148.4                                                                                                                                                                                                           | 1, 95                                                                                                                                                                                                                                                                                                                                                                                                                                                                                                                                                                                              | <0.001                                                                                                                                                                                                                                                                                                                                                  | 335                                      | 0.69                                       | 0.5                                             |
| -0.83+-0.083                            | 101.3                                                                                                                                                                                                           | 1, 250                                                                                                                                                                                                                                                                                                                                                                                                                                                                                                                                                                                             | <0.001                                                                                                                                                                                                                                                                                                                                                  |                                          |                                            | 0.05                                            |
|                                         | 1.9                                                                                                                                                                                                             | 1, 261                                                                                                                                                                                                                                                                                                                                                                                                                                                                                                                                                                                             | 0.16                                                                                                                                                                                                                                                                                                                                                    |                                          |                                            |                                                 |
| 0.49+-0.049                             | 100.5                                                                                                                                                                                                           | 1, 94                                                                                                                                                                                                                                                                                                                                                                                                                                                                                                                                                                                              | <0.001                                                                                                                                                                                                                                                                                                                                                  | 335                                      | 0.58                                       | 0.42                                            |
| -0.83+-0.083                            | 99.4                                                                                                                                                                                                            | 1, 249                                                                                                                                                                                                                                                                                                                                                                                                                                                                                                                                                                                             | <0.001                                                                                                                                                                                                                                                                                                                                                  |                                          |                                            | 0.05                                            |
|                                         | 1.4                                                                                                                                                                                                             | 1, 256                                                                                                                                                                                                                                                                                                                                                                                                                                                                                                                                                                                             | 0.24                                                                                                                                                                                                                                                                                                                                                    |                                          |                                            |                                                 |
| 0.37+-0.027                             | 187.2                                                                                                                                                                                                           | 1, 96                                                                                                                                                                                                                                                                                                                                                                                                                                                                                                                                                                                              | <0.001                                                                                                                                                                                                                                                                                                                                                  | 335                                      | 0.75                                       | 0.55                                            |
| -0.84+-0.083                            | 102.3                                                                                                                                                                                                           | 1, 252                                                                                                                                                                                                                                                                                                                                                                                                                                                                                                                                                                                             | <0.001                                                                                                                                                                                                                                                                                                                                                  |                                          |                                            | 0.05                                            |
|                                         | 2.3                                                                                                                                                                                                             | 1, 265                                                                                                                                                                                                                                                                                                                                                                                                                                                                                                                                                                                             | 0.13                                                                                                                                                                                                                                                                                                                                                    |                                          |                                            |                                                 |
| 0.41+-0.038                             | 111.9                                                                                                                                                                                                           | 1, 88                                                                                                                                                                                                                                                                                                                                                                                                                                                                                                                                                                                              | <0.001                                                                                                                                                                                                                                                                                                                                                  | 310                                      | 0.64                                       | 0.46                                            |
| 0.015+-0.005                            | 10.4                                                                                                                                                                                                            | 1, 288                                                                                                                                                                                                                                                                                                                                                                                                                                                                                                                                                                                             | <0.01                                                                                                                                                                                                                                                                                                                                                   |                                          | 0                                          | 0.01                                            |
| ,                                       | 0.1                                                                                                                                                                                                             | 1, 99                                                                                                                                                                                                                                                                                                                                                                                                                                                                                                                                                                                              | 0.81                                                                                                                                                                                                                                                                                                                                                    | 335                                      |                                            |                                                 |
| S.fus: -0.39+-0.48
S.mag: 0.28+-0.50 | 3.5                                                                                                                                                                                                             | 1, 266                                                                                                                                                                                                                                                                                                                                                                                                                                                                                                                                                                                             | 0.06                                                                                                                                                                                                                                                                                                                                                    |                                          | 0                                          | 0                                               |
| -0.0005+-0.0008                         | 0.4                                                                                                                                                                                                             | 1, 99                                                                                                                                                                                                                                                                                                                                                                                                                                                                                                                                                                                              | 0.54                                                                                                                                                                                                                                                                                                                                                    | 335                                      | 0                                          | 0                                               |
| i                                       | 0.8                                                                                                                                                                                                             | 1, 257                                                                                                                                                                                                                                                                                                                                                                                                                                                                                                                                                                                             | 0.37                                                                                                                                                                                                                                                                                                                                                    |                                          |                                            |                                                 |
| -0.14+-0.051                            | 7.4                                                                                                                                                                                                             | 1, 96                                                                                                                                                                                                                                                                                                                                                                                                                                                                                                                                                                                              | 0.01                                                                                                                                                                                                                                                                                                                                                    | 335                                      | 0.02                                       | 0.02                                            |
| i                                       | 1.6                                                                                                                                                                                                             | 1, 274                                                                                                                                                                                                                                                                                                                                                                                                                                                                                                                                                                                             | 0.21                                                                                                                                                                                                                                                                                                                                                    |                                          |                                            |                                                 |
|                                         | effect
0.43+-0.035
-0.83+-0.083
0.49+-0.049
-0.83+-0.083
0.37+-0.027
-0.84+-0.083
0.015+-0.005
0.5.fus: -0.39+-0.48
S.mag: 0.28+-0.50
2 -0.0005+-0.0008
3
-0.14+-0.051
3 | effect         F           0.43+-0.035         148.4           -0.83+-0.083         101.3           1.9         1.9           0.49+-0.049         100.5           -0.83+-0.083         99.4           -0.37+-0.027         187.2           -0.84+-0.083         102.3           2.3         2.3           0.41+-0.038         111.9           0.015+-0.005         10.4           9         0.1           S.fus: -0.39+-0.48         3.5           9         -0.0005+-0.0008         0.4           3         0.8           -0.0005+-0.001         7.4           5         -0.14+-0.051         7.4 | effectFDF $0.43+-0.035$ 148.41, 95 $-0.83+-0.083$ 101.31, 250 $1.9$ 1, 261 $0.49+-0.049$ 100.51, 94 $-0.83+-0.083$ 99.41, 249 $.0.37+-0.027$ 187.21, 96 $-0.84+-0.083$ 102.31, 252 $2.3$ 1, 265 $0.41+-0.038$ 111.91, 88 $0.015+-0.005$ 10.41, 288 $0.5$ fus: $-0.39+-0.48$ 3.51, 266 $2$ -0.0005+-0.00080.41, 99 $3$ -0.14+-0.0517.41, 96 $3.5$ 1, 257 | effectFDFP $0.43+-0.035$ 148.41,95

---

## Author Response (AR2)

LETTER TO THE EDITOR

Dear Dr. van der Meer,

Thank you for your comments and for your continued interest in our manuscript. We have incorporated your suggestions which included (1) new figures showing confidence intervals, and (2) a few clarifications of the text. In addition, we have done some minor fixes/updates of the manuscript.

Best regards,

Gustaf Granath and co-authors

**RESPONSE TO EDITOR COMMENTS**

I find that the s after composition in line 75 (annotated version) feels a bit weird.
RESPONSE: Changed.

In line 245 you use occasions, but perhaps time points is better.
RESPONSE: Changed.

Line 278 you have data from samples from a broad geographical area, I am not sure the current description works, although it is clear what you mean.
RESPONSE: Reformulated.

Line 322, the water film reduces CO2 transport and therefore results in less than maximum fractionation (or results in reduced fractionation).
RESPONSE: reformulated. "thus maintaining the waterfilm that hampers fractionation." -> "thus maintaining the waterfilm that results in reduced fractionation."

In line 343, enabling increased discrimination against (or increased fractionation).
RESPONSE: "enabling discrimination against" -> "enabling increased discrimination against"

Line 366, fewer precipitation collection stations, or weather stations?
RESPONSE: "fewer collection stations" -> "fewer precipitation collection stations"

Lines 382-384, to the best of my knowledge isotopically heavy water rains out first and with orographic uplift typically precipitation becomes more "depleted" with higher elevation not more "enriched" in 18O. Could the "amount" effect perhaps also play a role in your dataset?
RESPONSE: The text was referring to the study by Skrzypek et al. (2010) which did not find expected trends with altitude. It is correct that their non-significant result may be caused by variation in the precipitation amount at different elevations that can obscure altitude--d18O relationships. We have re-written this part.

[revised manuscript text omitted]